

# The potential impact of advanced footwear technology on the recent evolution of elite sprint performances

Joel Mason[1], Dominik Niedziela[2], Jean-Benoit Morin[3], Andreas Groll[2] and Astrid Zech[1]

[1] Department of Human Movement Science and Exercise Physiology, Friedrich Schiller University Jena, Jena, Germany
[2] Department of Statistics, TU Dortmund University, Dortmund, Germany
[3] Inter-University Laboratory of Human Movement Biology, University Jean Monnet Saint-Etienne, Saint-Etienne, France

Corresponding author
Joel Mason, joel.mason@uni-jena.de

## ABSTRACT

**Background:** Elite track and field sprint performances have reached a point of stability as we near the limits of human physiology, and further significant improvements may require technological intervention. Following the widely reported performance benefits of new advanced footwear technology (AFT) in road-running events, similar innovations have since been applied to sprint spikes in hope of providing similar performance enhancing benefits. However, it is not yet clear based on current evidence whether there have been subsequent improvements in sprint performance. Therefore, the aims of this study were to establish if there have been recent year-to-year improvements in the times of the annual top 100 and top 20 athletes in the men's and women's sprint events, and to establish if there is an association between the extensive use of AFT and potential recent improvements in sprint performances.

**Methods:** For the years 2016–19 and 2021–2022, the season best performances of the top 100 athletes in each sprint event were extracted from the World Athletics Top lists. Independent t-tests with Holm corrections were performed using the season's best performance of the top 100 and top 20 athletes in each year to identify significant differences between years for each sprint discipline. Following the classification of shoes worn by the top 20 athletes in each event during their annual best race (AFT or non-AFT), separate linear mixed-model regressions were performed to determine the influence of AFT on performance times.

**Results:** For the top 100 and top 20 athletes, there were no significant differences year-to-year in any sprint event prior to the release of AFT (2016–2019). There were significant differences between AFT years (2021 or 2022) and pre-AFT years (2016–2019) in eight out of 10 events. These differences ranged from a 0.40% improvement (men's 100 m) to a 1.52% improvement (women's 400 m hurdles). In the second analysis, multiple linear mixed model regressions revealed that the use of AFT was associated with improved performance in six out of ten events, including the men's and women's 100 m, women's 200 m, men's 110 m hurdles, women's 100 m hurdles and women's 400 m hurdles (estimate range: −0.037 – 0.521, $p$ = <0.001 – 0.021). Across both analyses, improvements were more pronounced in women's sprint events than men's sprint events.

**Conclusion:** Following a period of stability, there were significant improvements in most sprint events which may be partly explained by advances in footwear technology. These improvements appear to be mediated by event, sex and potentially level of athlete.

# INTRODUCTION

Track and field sprint events are among the most prominent and revered disciplines in the sporting world. The evolution of sprint performances over time reflects advancements in physiology and training methods, as well as technological innovation such as the introduction of synthetic tracks in the 1960s. Despite temporary regressions resulting from the implementation of automated timing and compulsory random drug testing, the 20th century was largely characterised by steady progress in elite sprint performances (*Haake, Foster & James, 2014*; *Lippi et al., 2008*). Following a century of progress, sprint times have now somewhat plateaued since the 1990s across most elite sprint disciplines as performances have approached their asymptotic limits (*Berthelot et al., 2010*, *2015*; *Weiss et al., 2016*; *Ganse & Degens, 2021*). This plateau is particularly prominent in the women's events. One model incorporating performances from 1896–2008 indicates that no meaningful progression has occurred in four out of five women's sprint events since 1994 (*Berthelot et al., 2010*), which may be partially explained by the introduction of routine performance enhancing drug testing (*Haake, Foster & James, 2014*). Similar performance stagnations have been observed across field events and long-distance running events for both sexes (*Berthelot et al., 2010*; *Haake, James & Foster, 2015*), adding credence to the wider notion that we are approaching the limits of human physiology (*Berthelot et al., 2008*; *Nevill & Whyte, 2005*; *Haugen, Tønnessen & Seiler, 2015*).

In order to further substantially improve human performances, exogenous measures to overcome the limits of our physiology may be required, including artificial conditions and new technologies (*Marck et al., 2017*). For road running events, the recent introduction of advanced footwear technology (AFT, *Frederick, 2022*) has marked a new era in long-distance running performance, headlined by new world records in every distance from 5-km to the marathon for both men and women. AFT's combination of "lightweight resilient midsole foams with rigid moderators and pronounced rocker profiles in the sole" (*Frederick, 2022*) has been demonstrated to improve the metabolic cost of running compared to conventional marathon shoes (*Hoogkamer et al., 2018*). Analyses of the annual top 100 times worldwide across all road-running distances following the introduction of AFT confirm the paradigm shift, indicating that road-racing times have improved by 1–3% since their release (*Senefeld et al., 2021*; *Rodrigo-Carranza et al., 2021*; *Bermon et al., 2021*; *Rodrigo-Carranza et al., 2022*). Subsequent to this resounding success, similar innovative upgrades have since been introduced in track spikes for both sprint and middle-distance disciplines, with the ultimate ambition of inducing similar

performance-enhancing effects. So-called *superspikes* use an analogous approach of a plated midsole (often carbon fibre or nylon) combined with a thick midsole of foam (or pods of air), which is a clear departure from preceding sprint spike designs emphasising slim midsoles to minimise weight. Carbon fibre plates are not a recent introduction to sprint spikes, and there is evidence of how this longitudinal bending stiffness may influence both acceleration and maximal velocity (*Stefanyshyn & Fusco, 2004*; *Smith et al., 2016*; *Willwacher et al., 2016*). However, there is no publicly available evidence demonstrating how changes in midsole material and midsole thickness may influence sprinting when paired with the increased longitudinal bending stiffness provided by a plated sole. Therefore, precisely how this new generation of spikes interacts with the biomechanical and metabolic determinants of sprinting to potentially augment performance remains unclear (*Healey et al., 2022*). Further, how these potential benefits may vary according to sex and ability level, both factors suggested to mediate the performance enhancing effects of AFT on long-distance running performance (*Knopp et al., 2023*; *Senefeld et al., 2021*; *Bermon et al., 2021*), is also unknown.

Although high-quality evidence for the mechanisms and associated performance-enhancing effects is currently lacking, AFT sprint spikes have been widely adopted by both recreational and elite sprinters, and there are preliminary indications of a potential shift in elite sprint performances. Since the introduction of AFT to sprinting in 2020, there have been world records set in the men's and women's 400 m hurdles, women's indoor 400 m and world junior records in the men's 100 and 200 m. Further, although only 50% of gold medals in throwing events at the Tokyo Olympics exceeded the performance from the Rio Olympics five years earlier, 90% of sprinting gold medals exceeded the performances from Rio. Additionally, there is plausible theory underlying an AFT-induced improvement in sprint times (*Healey et al., 2022*). However despite these factors, there has yet to be a systematic appraisal of the influence of AFT on elite sprint performances, with only a pre-print available which provides no link between AFT and performance improvements (*Willwacher et al., 2023*). Therefore, the primary aims of this study were (1) to establish if there have been recent year-to-year improvements in the annual top 100 and top 20 athletes of men's and women's sprint events, and (2) to establish if there is an association between the introduction of AFT and the potential recent improvements in sprint performances in each event. We hypothesised that recent improvements in sprint times will be at least partially be explained by the use of AFT.

## MATERIALS AND METHODS

All procedures adhered to the Declaration of Helsinki and were approved by the ethics committee of the Friedrich Schiller University Jena (approval number: FSV 23/057). Due to all analysis involving data available in the public domain, informed consent was not required.

### Database search and data selection

For the years 2016–19 and 2021–2022, the season best performances of the top 100 athletes in each sprint event were extracted from the World Athletics Top lists (*World Athletics,*

 

*2023*; accessed January 2023), including the men's and women's 100, 200, 400, 400 m hurdles, women's 100 m hurdles and men's 110 m hurdles. Only one performance per athlete was recorded, and only wind-legal times recorded electronically in an outdoor competition were included. The year 2020 was also excluded due to significant pandemic-induced interruptions to training and competition opportunities, including the postponement of the 2020 Olympic Games. We selected 2016 as a cut-off point to capture the most recent evolution in sprint performances, in line with time periods used by previous studies characterising the influence of AFT on road-racing times (*Rodrigo-Carranza et al., 2021*, *2022*). Data from 2010 onwards is included as a supplementary file (Supplementary File 2), which, alongside of the results of *Willwacher et al. (2023)*, indicates that altering the time period of our study has no bearing on our analysis and subsequent findings.

## Definition and identification of AFT

For the top 20 performers in each event in the years 2021 and 2022, two investigators independently identified the shoes worn in each athlete's season best race in order to classify the footwear worn as either AFT or non-AFT. As for *Bermon et al. (2021)*, the identification of the footwear of the top 100 athletes was not feasible due to limited availability of information. Identification of spikes used in each race was completed through media content, including race footage or photos from athlete and event social media, YouTube, or other official event photography services available online. Any disagreement was resolved by consensus with the remaining authors. Previous studies have used the same method to identify AFT use in elite road-race athletes (*Senefeld et al., 2021*, *Rodrigo-Carranza et al., 2021*; *Bermon et al., 2021*).

AFT was defined as per *Healey et al. (2022)*, whereby a superspike incorporates "a combination of lightweight, compliant and resilient foams (and/or air pods) with a stiff (nylon, PEBA, carbon-fiber) plate". Therefore, spikes which contained only a stiff plate or only a thick midsole of innovative foam without the presence of the other were not classified as AFT. Eligibility of models was assessed through manufacturer details of shoe composition available online.

## Data analysis and statistics

Multiple one-sided independent t-tests with Holm correction (*Holm, 1979*) were performed using the season's best performance of the top 100 athletes in each year to identify significant differences between the years 2016, 2017, 2018, 2019, 2021 and 2022 in each event.

To verify the normal distribution assumption of our data for the t-test, visual analyses with kernel density estimations were completed. A Levene's-Test was also conducted to test for unequal variances within the events (*Levene, 1960*). As the normality assumption appears to be somewhat critical in some events, particularly because the underlying top 20 or top 100 performance variables are cut off at the upper tails, we additionally performed Wilcoxon–Mann–Whitney tests in order to validate the findings from our t-test analysis. This approach tested for the null hypothesis that it is equally likely that a value chosen at

random from one year is greater or less than a value chosen at random from another year's population), and the findings can be found in Supplementary File 3. Identical analysis was performed using the season best performances of solely the top 20 athletes each year to provide the basis for the regression analyses assessing the impact of AFT on performances.

Following the identification and classification of shoes worn by the top 20 athletes in each event during their season best race, separate linear mixed-model regressions were performed for each event to determine the influence of AFT on performance times. Use of AFT (or not) and year were used as fixed effects, and participant ID used as random effect predictors. For the linear mixed models, the normality assumption is also implicitly relevant. However, on the basis of a careful goodness-of-fit analysis using residual and QQ-plots, we found no mentionable violations here for all fitted models.

All data analysis and visualisations were completed in R (*R Core Team, 2023*). The t-test analyses were performed using the pairwise.t.test function from the base R package. For the mixed effects regression analyses, the packages lme4 and lmerTest were applied. For the visualisations, the packages dplyr and ggplot2 were employed. Significance for all analyses was set at $p < 0.05$, and Cohen's *d* to calculate effect size, with values of <0.5, 0.5–0.79 and >0.80 considered small, medium and large respectively (*Cohen, 1992*).

## RESULTS

### Comparison of the annual top 100 sprint performances in each sprint event

Table 1 and Figs. 1–3 demonstrate the changes in the season best performances of the top 100 men and women in each sprint event between the years 2016–2022. For the pre-AFT period (2016–2019), no meaningful changes and no significant improvements were detected *via* t-test comparisons with Holm correction in the top 100 times in all sprint events for both sexes (Table 1). The sprint times of the AFT era years (2021 or 2022) were significantly faster compared to sprint times from the pre-AFT era in seven sprint events (women's 100, 200, 400, 100 m hurdles and men's 100, 200 and 110 m hurdles), with significant improvements ranging from 0.40% (men's 100 m) to 0.90% (women's 100 m) (Table 2). For the women's 100 m, women's 400 m and men's 110 m hurdles, the year 2022 was significantly faster than all pre-AFT years.

Figures 1–3 displays the raw data together with boxplots and kernel density estimates for all ten sprint events. In most events, the distributions of the year 2021 (magenta) and 2022 (blue) are clearly shifted down in comparison to earlier years, indicating an improvement in times.

### Comparison of the annual top 20 sprint performances in each sprint event

Table 3 demonstrates the changes in the season best performances of the top 20 men and women in each sprint event between the years 2016–2022. For the pre-AFT period (2016–2019), no meaningful changes and no significant improvements were detected *via*

**Table 1 Annual best performances of the top 100 athletes in each sprint event.** Times listed as seconds (mean ± SD).

| | Pre-AFT period | | | | AFT period | |
|---|---|---|---|---|---|---|
| Event (m) | 2016 | 2017 | 2018 | 2019 | 2021 | 2022 |
| **W100** | 11.13 ± 0.14[#] | 11.13 ± 0.14[#] | 11.13 ± 0.11[#] | 11.14 ± 0.12[*#] | 11.09 ± 0.14 | 11.04 ± 0.13 |
| **M100** | 10.04 ± 0.08 | 10.07 ± 0.08[#] | 10.06 ± 0.08[#] | 10.06 ± 0.08[#] | 10.04 ± 0.10 | 10.02 ± 0.08 |
| **W200** | 22.68 ± 0.27 | 22.73 ± 0.28[*] | 22.69 ± 0.26[*] | 22.77 ± 0.30[*#] | 22.63 ± 0.35 | 22.57 ± 0.33 |
| **M200** | 20.25 ± 0.19 | 20.29 ± 0.16[#] | 20.25 ± 0.20 | 20.26 ± 0.21[#] | 20.25 ± 0.20 | 20.18 ± 0.22 |
| **W400** | 51.41 ± 0.68[*#] | 51.46 ± 0.68[*#] | 51.32 ± 0.72[*#] | 51.39 ± 0.73[*#] | 50.99 ± 0.75 | 51.07 ± 0.65 |
| **M400** | 45.18 ± 0.50 | 45.13 ± 0.46 | 45.10 ± 0.47 | 45.17 ± 0.53 | 45.15 ± 0.46 | 45.08 ± 0.43 |
| **W100H** | 12.87 ± 0.16 | 12.91 ± 0.19[*#] | 12.89 ± 0.19[*] | 12.88 ± 0.20[*] | 12.83 ± 0.18 | 12.80 ± 0.22 |
| **M110H** | 13.43 ± 0.16[#] | 13.44 ± 0.17[*#] | 13.48 ± 0.16[*#] | 13.46 ± 0.16[*#] | 13.38 ± 0.17 | 13.36 ± 0.16 |
| **W400H** | 55.69 ± 0.91 | 55.78 ± 1.07 | 55.94 ± 1.00 | 55.90 ± 1.05 | 55.62 ± 1.19 | 55.55 ± 1.16 |
| **M400H** | 49.23 ± 0.53 | 49.22 ± 0.51 | 49.20 ± 0.59 | 49.25 ± 0.59 | 49.14 ± 0.80 | 49.07 ± 0.70 |

Notes:
[*] Significantly slower than 2021 (*via* t-test comparison).
[#] Significantly slower than 2022 (*via* t-test comparison).
AFT, advanced footwear technology; W, women's; M, men's; H, hurdles; SD, standard deviation.

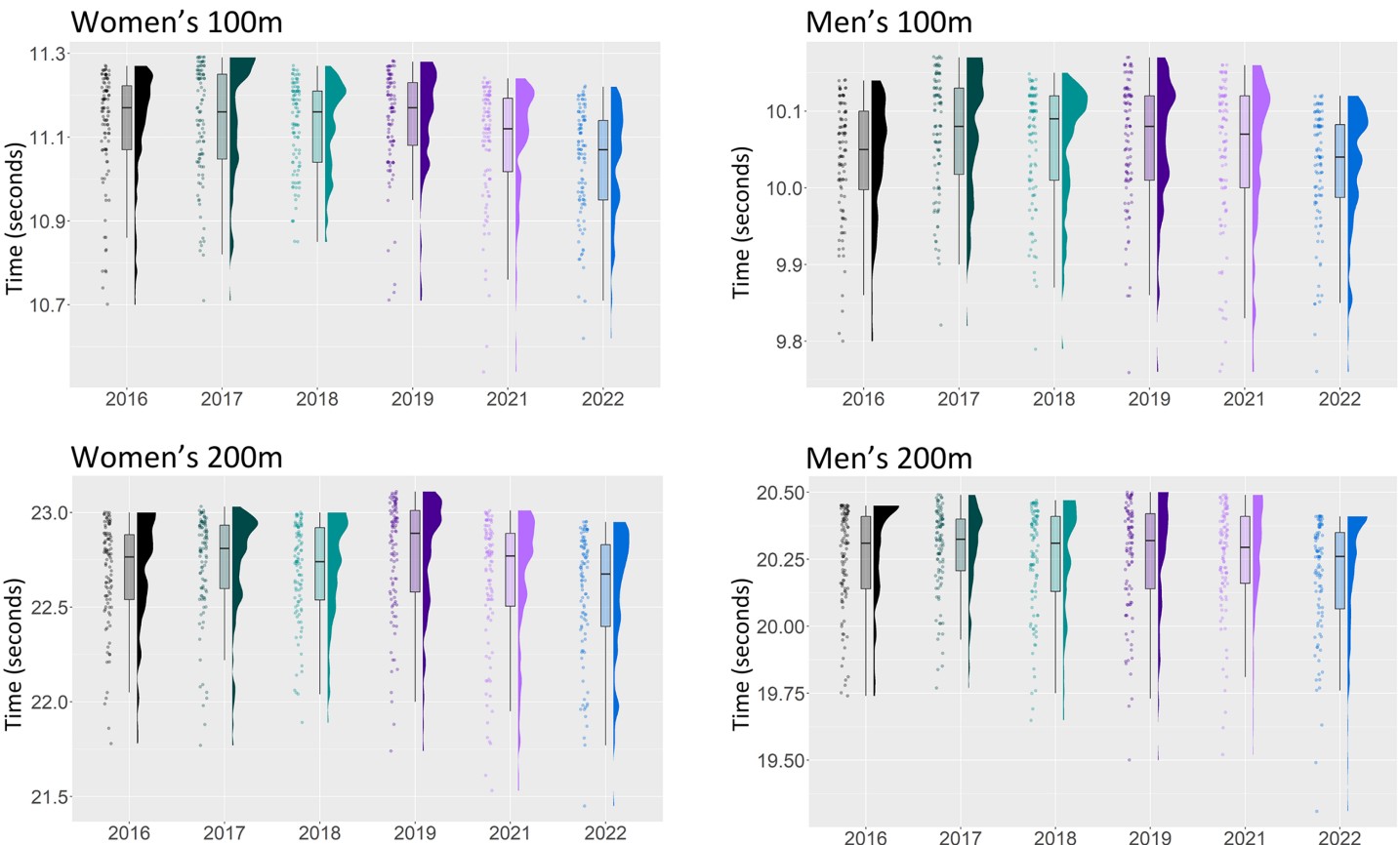

**Figure 1 Season best performances of the top 100 athletes from the years 2016–2022 in the men's and women's 100 and 200 m.**

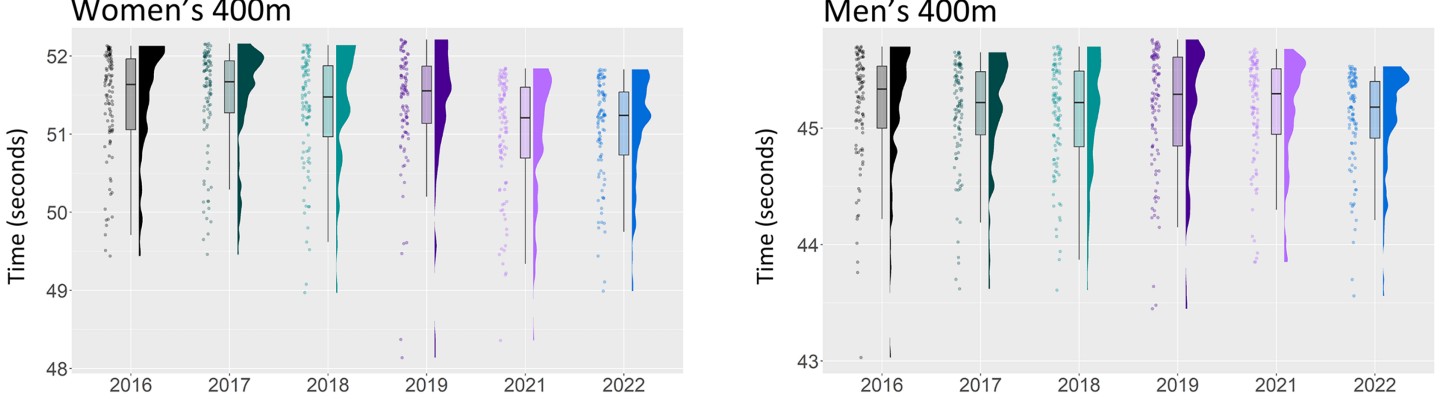

**Figure 2 Season best performances of the top 100 athletes from the years 2016–2022 in the men's and women's hurdles.**

**Figure 3 Season best performances of the top 100 athletes from the years 2016–2022 in the men's and women's 400 m.**

t-test comparisons in the top 20 times in all sprint events for both sexes (Tables 3 and 4). The sprint times of the AFT era years (2021 or 2022) were significantly faster compared to sprint times from the pre-AFT era in eight sprint events (women's 100, 200, 400, 100 m

**Table 2 Overview of significant year-to-year differences in the annual top 100 sprint performances.**

| Event and comparison | Performances (s) | Δ | p-value | Effect size |
|---|---|---|---|---|
| **Women's 100 m** | | | | |
| 2019–2021 | 11.14 ± 0.12 vs 11.09 ± 0.14 | 0.45% | 0.027 | −0.387 |
| 2016–2022 | 11.13 ± 0.14 vs 11.04 ± 0.13 | 0.81% | <0.001 | −0.662 |
| 2017–2022 | 11.13 ± 0.14 vs 11.04 ± 0.13 | 0.81% | <0.001 | −0.710 |
| 2018–2022 | 11.13 ± 0.11 vs 11.04 ± 0.13 | 0.81% | <0.001 | −0.661 |
| 2019–2022 | 11.14 ± 0.12 vs 11.04 ± 0.13 | 0.90% | <0.001 | −0.760 |
| 2021–2022 | 11.09 ± 0.14 vs 11.04 ± 0.13 | 0.45% | 0.033 | −0.373 |
| **Men's 100 m** | | | | |
| 2017–2022 | 10.07 ± 0.08 vs 10.02 ± 0.08 | 0.50% | <0.001 | −0.553 |
| 2018–2022 | 10.06 ± 0.08 vs 10.02 ± 0.08 | 0.40% | 0.034 | −0.391 |
| 2019–2022 | 10.06 ± 0.08 vs 10.02 ± 0.08 | 0.40% | 0.012 | −0.430 |
| **Women's 200 m** | | | | |
| 2019–2021 | 22.77 ± 0.30 vs 22.63 ± 0.35 | 0.62% | 0.005 | −0.470 |
| 2017–2022 | 22.73 ± 0.28 vs 22.57 ± 0.33 | 0.71% | 0.002 | 0.517 |
| 2018–2022 | 22.69 ± 0.26 vs 22.57 ± 0.33 | 0.53% | 0.028 | −0.395 |
| 2019–2022 | 22.77 ± 0.30 vs 22.57 ± 0.33 | 0.88% | <0.001 | −0.666 |
| **Men's 200 m** | | | | |
| 2017–2022 | 20.29 ± 0.16 vs 20.18 ± 0.22 | 0.54% | 0.002 | −0.525 |
| 2019–2022 | 20.26 ± 0.21 vs 20.18 ± 0.22 | 0.40% | 0.045 | −0.384 |
| **Women's 400 m** | | | | |
| 2016–2021 | 51.41 ± 0.68 vs 50.99 ± 0.75 | 0.82% | <0.001 | −0.589 |
| 2017–2021 | 51.46 ± 0.68 vs 50.99 ± 0.75 | 0.92% | <0.001 | −0.654 |
| 2018–2021 | 51.32 ± 0.72 vs 50.99 ± 0.75 | 0.65% | 0.004 | −0.462 |
| 2019–2021 | 51.39 ± 0.73 vs 50.99 ± 0.75 | 0.78% | <0.001 | −0.553 |
| 2016–2022 | 51.41 ± 0.68 vs 51.07 ± 0.65 | 0.67% | 0.003 | −0.476 |
| 2017–2022 | 51.46 ± 0.68 vs 51.07 ± 0.65 | 0.76% | 0.001 | −0.541 |
| 2018–2022 | 51.32 ± 0.72 vs 51.07 ± 0.65 | 0.49% | 0.046 | −0.349 |
| 2019–2022 | 51.39 ± 0.73 vs 51.07 ± 0.65 | 0.62% | 0.006 | −0.440 |
| **Women's 100 m H** | | | | |
| 2017–2021 | 12.91 ± 0.19 vs 12.83 ± 0.18 | 0.62% | 0.028 | −0.397 |
| 2017–2022 | 12.91 ± 0.19 vs 12.80 ± 0.22 | 0.86% | 0.001 | −0.541 |
| 2018–2022 | 12.89 ± 0.19 vs 12.80 ± 0.22 | 0.70% | 0.001 | −0.455 |
| 2019–2022 | 12.88 ± 0.20 vs 12.80 ± 0.22 | 0.62% | 0.027 | −0.401 |
| **Men's 110 m H** | | | | |
| 2017–2021 | 13.44 ± 0.17 vs 13.38 ± 0.17 | 0.45% | 0.043 | −0.358 |
| 2018–2021 | 13.48 ± 0.16 vs 13.38 ± 0.17 | 0.74% | <0.001 | −0.570 |
| 2019–2021 | 13.46 ± 0.16 vs 13.38 ± 0.17 | 0.59% | <0.001 | −0.436 |
| 2016–2022 | 13.43 ± 0.16 vs 13.36 ± 0.16 | 0.52% | 0.014 | −0.414 |
| 2017–2022 | 13.44 ± 0.17 vs 13.36 ± 0.16 | 0.60% | 0.003 | −0.486 |
| 2018–2022 | 13.48 ± 0.16 vs 13.36 ± 0.16 | 0.89% | <0.001 | −0.698 |
| 2019–2022 | 13.46 ± 0.16 vs 13.36 ± 0.16 | 0.75% | <0.001 | −0.564 |

Note:
M, metres; H, hurdles; Δ, percentage change.

**Table 3 Annual best performances of the top 20 athletes in each sprint event, reported in seconds (mean ± SD).**

| | Pre-AFT period | | | | AFT period | |
|---|---|---|---|---|---|---|
| Event (m) | 2016 | 2017 | 2018 | 2019 | 2021 | 2022 |
| **W100** | 10.90 ± 0.11 | 10.92 ± 0.08* | 10.96 ± 0.05*# | 10.96 ± 0.11*# | 10.86 ± 0.12 | 10.84 ± 0.08 |
| **M100** | 9.92 ± 0.04 | 9.95 ± 0.03*# | 9.93 ± 0.04# | 9.94 ± 0.04*# | 9.89 ± 0.06 | 9.90 ± 0.05 |
| **W200** | 22.25 ± 0.21*# | 22.29 ± 0.25*# | 22.29 ± 0.18*# | 22.28 ± 0.23*# | 22.02 ± 0.23 | 22.02 ± 0.20 |
| **M200** | 19.94 ± 0.12 | 20.02 ± 0.12# | 19.91 ± 12 | 19.92 ± 0.15 | 19.94 ± 0.18 | 19.83 ± 0.18 |
| **W400** | 50.27 ± 0.47# | 50.27 ± 0.25# | 50.15 ± 0.52 | 50.30 ± 0.84# | 49.74 ± 0.48 | 50.00 ± 0.43 |
| **M400** | 44.38 ± 0.45 | 44.37 ± 0.37 | 44.35 ± 0.30 | 44.29 ± 0.37 | 44.39 ± 0.33 | 44.36 ± 0.32 |
| **W100H** | 12.61 ± 0.14# | 12.59 ± 0.11# | 12.60 ± 0.13# | 12.57 ± 0.12# | 12.53 ± 0.10 | 12.44 ± 0.13 |
| **M110H** | 13.17 ± 0.09 | 13.17 ± 0.12 | 13.23 ± 0.10*# | 13.20 ± 0.11# | 13.12 ± 0.10 | 13.10 ± 0.10 |
| **W400H** | 54.21 ± 0.52 | 53.98 ± 0.70 | 54.47 ± 0.69# | 54.25 ± 0.88 | 53.76 ± 1.12 | 53.65 ± 0.90 |
| **M400H** | 48.48 ± 0.36 | 48.37 ± 0.23 | 48.25 ± 0.56 | 48.32 ± 0.59 | 47.89 ± 0.84 | 47.93 ± 0.67 |

**Notes:**
* Significantly slower than 2021 (*via* t-test comparison).
# Significantly slower than 2022 (*via* t-test comparison).
AFT, advanced footwear technology; W, women's; M, men's; H, hurdles; SD, standard deviation.

**Table 4 Overview of significant year-to-year differences in the annual top 20 sprint performances (according to t-test comparison).**

| Event and comparison | Performances (s) | Δ | p-value | Effect size |
|---|---|---|---|---|
| **Women's 100 m** | | | | |
| 2018–2021 | 10.96 ± 0.05 *vs* 10.86 ± 0.12 | 0.92% | 0.013 | −0.920 |
| 2019–2021 | 10.96 ± 0.11 *vs* 10.86 ± 0.12 | 0.92% | 0.013 | −0.924 |
| 2017–2022 | 10.92 ± 0.08 *vs* 10.84 ± 0.08 | 0.74% | 0.046 | −0.781 |
| 2018–2022 | 10.96 ± 0.05 *vs* 10.84 ± 0.08 | 1.10% | 0.001 | −1.143 |
| 2019–2022 | 10.96 ± 0.11 *vs* 10.84 ± 0.08 | 1.10% | 0.001 | −1.148 |
| **Men's 100 m** | | | | |
| 2017–2021 | 9.95 ± 0.03 *vs* 9.89 ± 0.06 | 0.60% | 0.005 | −1.059 |
| 2019–2021 | 9.94 ± 0.04 *vs* 9.89 ± 0.06 | 0.50% | 0.039 | −0.842 |
| 2017–2022 | 9.95 ± 0.03 *vs* 9.90 ± 0.05 | 0.50% | 0.035 | −0.860 |
| **Women's 200 m** | | | | |
| 2016–2021 | 22.25 ± 0.21 *vs* 22.02 ± 0.23 | 1.04% | 0.006 | −0.917 |
| 2017–2021 | 22.29 ± 0.25 *vs* 22.02 ± 0.23 | 1.22% | 0.001 | −1.073 |
| 2018–2021 | 22.29 ± 0.18 *vs* 22.02 ± 0.23 | 1.29% | 0.001 | −1.065 |
| 2019–2021 | 22.28 ± 0.23 *vs* 22.02 ± 0.23 | 1.17% | 0.001 | −1.063 |
| 2016–2022 | 22.25 ± 0.21 *vs* 22.02 ± 0.20 | 1.04% | 0.004 | −0.949 |
| 2017–2022 | 22.29 ± 0.25 *vs* 22.02 ± 0.20 | 1.22% | 0.001 | −1.106 |
| 2018–2022 | 22.29 ± 0.18 *vs* 22.02 ± 0.20 | 1.29% | 0.001 | −1.098 |
| 2019–2022 | 22.28 ± 0.23 *vs* 22.02 ± 0.20 | 1.17% | 0.001 | −1.096 |
| **Men's 200 m** | | | | |
| 2017–2022 | 20.02 ± 0.12 *vs* 19.83 ± 0.18 | 0.95% | <0.001 | −1.255 |
| **Women's 400 m** | | | | |
| 2016–2021 | 50.27 ± 0.47 *vs* 49.74 ± 0.48 | 1.06% | 0.020 | −0.925 |
| 2017–2021 | 50.27 ± 0.25 *vs* 49.74 ± 0.48 | 1.06% | 0.020 | −0.921 |

(Continued)

| Table 4 (continued) | | | | |
|---|---|---|---|---|
| Event and comparison | Performances (s) | Δ | p-value | Effect size |
| 2019–2021 | 50.30 ± 0.84 vs 49.74 ± 0.48 | 1.12% | 0.014 | −0.967 |
| **Women's 100 m H** | | | | |
| 2016–2022 | 12.61 ± 0.14 vs 12.44 ± 0.13 | 1.36% | <0.001 | −1.279 |
| 2017–2022 | 12.59 ± 0.11 vs 12.44 ± 0.13 | 1.20% | 0.001 | −1.159 |
| 2018–2022 | 12.60 ± 0.13 vs 12.44 ± 0.13 | 1.28% | <0.001 | −1.216 |
| 2019–2022 | 12.57 ± 0.12 vs 12.44 ± 0.13 | 1.04% | 0.004 | −1.010 |
| **Men's 110 m H** | | | | |
| 2018–2021 | 13.23 ± 0.10 vs 13.12 ± 0.10 | 0.83% | 0.007 | −1.009 |
| 2018–2022 | 13.23 ± 0.10 vs 13.10 ± 0.10 | 0.98% | 0.002 | −1.112 |
| 2019–2022 | 13.20 ± 0.11 vs 13.10 ± 0.10 | 0.76% | 0.026 | −0.870 |
| **Women's 400 m H** | | | | |
| 2018–2022 | 54.47 ± 0.69 vs 53.65 ± 0.90 | 1.52% | 0.015 | −0.960 |

Note:
M, metres; H, hurdles; Δ, percentage change.

Table 5 Number of top 20 athletes wearing AFT, non-AFT and unidentifiable spikes in the years 2021 and 2022 for each sprint event.

| Event (m) | 2021 | | | 2022 | | |
|---|---|---|---|---|---|---|
| | Non-AFT | AFT | Unidentified | Non-AFT | AFT | Unidentified |
| **W100** | 9 | 11 | 0 | 1 | 18 | 1 |
| **M100** | 8 | 12 | 0 | 2 | 18 | 0 |
| **W200** | 10 | 9 | 1 | 1 | 19 | 0 |
| **M200** | 13 | 7 | 0 | 3 | 16 | 1 |
| **W400** | 8 | 11 | 1 | 0 | 19 | 1 |
| **M400** | 11 | 9 | 0 | 3 | 17 | 0 |
| **W100H** | 15 | 5 | 0 | 1 | 17 | 2 |
| **M110H** | 15 | 4 | 1 | 4 | 16 | 0 |
| **W400H** | 9 | 11 | 0 | 0 | 20 | 0 |
| **M400H** | 14 | 6 | 0 | 3 | 16 | 1 |

Note:
AFT, advanced footwear technology; W, women's; M, men's; H, hurdles.

hurdles and 400 m hurdles and men's 100, 200 and 110 m hurdles), with significant improvements ranging from 0.50% (men's 100 m) to 1.52 % (women's 400 m hurdles) (Table 4). For the women's 200 m and women's 100 m hurdles, the year 2022 was significantly faster than all pre-AFT years.

## The influence of AFT on recent sprint performances

A total of 97.75% of shoes worn by the top 20 athletes of 2021 and 2022 in their season best performance were able to be identified *via* media content (Table 5).

According to the mixed effects models, the use of AFT significantly improved performance in six out of ten events, including the men's and women's 100 m, women's

**Table 6 The estimated regression effect of AFT usage on performance times in each sprint event according to linear mixed effects models.**

| | Fixed effects | | | | | | Random effects | | | |
| | Use of AFT | | | Year | | | Athlete | | Residual | |
| Event (m) | Estimate | Error | *p*-value | Estimate | Error | *p*-value | Variance | SD | Variance | SD |
|---|---|---|---|---|---|---|---|---|---|---|
| W100 | −0.106 | 0.027 | <0.001* | 0.004 | 0.006 | 0.493 | 0.004 | 0.060 | 0.005 | 0.072 |
| M100 | −0.053 | 0.016 | 0.001* | 0.001 | 0.003 | 0.698 | 0.001 | 0.030 | 0.002 | 0.043 |
| W200 | −0.149 | 0.064 | 0.021* | −0.031 | 0.013 | 0.022* | 0.015 | 0.123 | 0.032 | 0.179 |
| M200 | −0.037 | 0.045 | 0.411 | −0.014 | 0.009 | 0.100 | 0.006 | 0.078 | 0.016 | 0.127 |
| W400 | −0.084 | 0.171 | 0.623 | −0.067 | 0.035 | 0.057 | 0.111 | 0.333 | 0.173 | 0.416 |
| M400 | −0.190 | 0.104 | 0.070 | 0.030 | 0.020 | 0.139 | 0.027 | 0.165 | 0.093 | 0.305 |
| W100H | −0.093 | 0.034 | 0.008* | 0.017 | 0.007 | 0.014* | 0.004 | 0.064 | 0.009 | 0.097 |
| M110H | −0.087 | 0.030 | 0.005* | −0.003 | 0.005 | 0.621 | 0.004 | 0.060 | 0.007 | 0.086 |
| W400H | −0.521 | 0.216 | 0.018* | −0.020 | 0.045 | 0.589 | 0.285 | 0.534 | 0.303 | 0.551 |
| M400H | −0.085 | 0.155 | 0.586 | −0.081 | 0.030 | 0.007* | 0.128 | 0.358 | 0.170 | 0.413 |

Notes:
* Statistical significance ($p = < 0.05$).
AFT, advanced footwear technology; W, women's; M, men's; H, hurdles; SD, standard deviation.

200 m, men's 110 m hurdles, women's 100 m hurdles and women's 400 m hurdles (Table 6).

## DISCUSSION

We sought to identify whether there have been recent changes in the annual top sprint performances, and to subsequently evaluate the influence of AFT on elite sprint times. Our key findings include that: (1) following a plateau in performances in all sprint events between 2016–2019, statistically significant and specific improvements were identified in most sprint disciplines which coincided with widespread adoption of AFT in 2021 and particularly 2022, (2) these significant improvements ranged from 0.40–1.52%, and were typically more pronounced in women's events than men's events, (3) the use of AFT may partially explain these recent improvements in sprint times, with a significant relationship identified in six out of ten events.

This study provides the first peer-reviewed evidence suggesting that performances in some sprint events have significantly improved following a period of stability, and that this improvement has been at least partially driven by the widespread adoption of AFT. Although the changes in performance were less substantial, less consistent and less unanimous as the AFT-induced performance improvements in road-running events with longer distances (*Rodrigo-Carranza et al., 2022, 2021; Bermon et al., 2021*), our results provide initial evidence that along with the technological innovation there is meaningful advancement in sprint performances. This finding is also in line with a recent pre-print using a similar approach to characterise improvements in sprint times between 2010–2022 (*Willwacher et al., 2023*).

A key cornerstone of our findings is that between 2016–2019, there were no significant differences in the season best performances of the top 100 or top 20 athletes in any of the

sprint events, supporting the notion that performances had reached a plateau as we likely near the limits of human physiology (*Berthelot et al., 2008*; *Nevill & Whyte, 2005*; *Haugen, Tønnessen & Seiler, 2015*). This adds substantial weight to our finding that AFT is likely a factor explaining recent performance improvements. For example, in the annual top 100 performances in the women's 100 m prior to the release of AFT (2016–2019), the average performance was stable between 11.13–11.14 s, with an average year-to-year variation of less than 0.1%, which underlines the significance of the 0.90% improvement in 2022 compared to 2019.

There are a number of candidate mechanisms which potentially accrue and interact to underpin the performance benefits of AFT observed in the current study. Given that carbon fibre plates in isolation have existed in sprint spikes for an extended period of time, the presence of a stiff plate alone is likely insufficient to explain the significant improvements in sprint times, and instead it is more likely that innovative midsole materials and geometry are the key drivers alongside longitudinal bending stiffness. For example, new foams such as polyether block amide demonstrate far superior energy restitution than traditional midsoles made of ethylene–vinyl acetate (*Hoogkamer et al., 2018*). In addition to the composition of the midsole, the increased thickness/height of the midsole (and its spatial distribution beneath the foot) in the new generation of spikes compared to traditionally minimal racing spikes potentially provides several advantages. An increase in the midsole thickness, which is capped at 20 mm by World Athletics regulations (*World Athletics, 2021*), may create more beneficial lever arms, potentially creating favourable shifts in ratio of force during acceleration towards horizontal reaction ground force orientation, which is a central determinant of sprint performance (*Morin, Edouard & Samozino, 2011*; *Rabita et al., 2015*). Changes in both shank position and dorsiflexion range of movement, both of which may be achieved *via* a higher midsole stack height, have been recently linked with better ratio of force during acceleration (*King et al., 2022*).

Further, an increase in midsole thickness may result in between a 1–3% increase in overall limb length and enhance stride length, the consequences of which are increasingly studied in the context of athletes with transtibial amputations. Although the topic is currently keenly debated (*Taboga et al., 2020*; *Beck, Taboga & Grabowski, 2022*; *Zhang-Lea et al., 2023*; *Weyand et al., 2022.*), there is evidence of an association between longer leg length and faster maximal velocity (*Weyand et al., 2022*). In the world's best transtibial amputation 400 m runner, reducing limb length by 5 cm produced a substantial drop in maximal treadmill velocity from 11.4 to 10.9 m/s (*Weyand et al., 2022*), leading to substantial projected and actual reductions in race performance. Although reduced leg length in amputee athletes resulting in slower speed does not guarantee that increasing leg length results in higher speed in able-bodied athletes, there is also evidence from non-amputee athletes that longer leg lengths may be particularly beneficial for longer sprinting (*i.e.*, 400 m) (*Weyand & Davis, 2005*; *Tomita et al., 2020*). These factors, combined with improvements in running economy (*Hoogkamer et al., 2018*), which are increasingly valuable in distances over 100 m, potentially explain some of the performance enhancing effects of AFT observed in this study. It should be noted that these mechanisms

remain primarily speculative at this stage, based on studies which do not directly investigate midsole thickness. Importantly, an increase in midsole thickness alone is not sufficient to improve running economy over longer distances (*Barrons, Wannop & Stefanyshyn, 2023*), indicating that if midsole thickness is indeed involved in performance enhancements, then it likely acts in concert with other components of the footwear, including the longitudinal bending stiffness. A key example which further demonstrates the uncertainty of the mechanisms is that AFT potentially also creates a *less* beneficial lever arm, considering that for a constant hip torque, a longer effective leg length will result in a smaller propulsive force. Therefore, future studies should seek to clarify the precise mechanisms through which AFT ultimately contributes to enhanced sprint performance.

This study also demonstrated *via* two unique methods that women's sprint events have undergone more substantial and more widespread recent improvements than male events, and that AFT had a greater impact on women's performances than men's performances. This is consistent with road-running research indicating that women benefited more from AFT than men (*Bermon et al., 2021*), including the findings that AFT improved marathon finishing time by 0.8% for males and by 1.6% for females in a subsample of marathon finishers (*Senefeld et al., 2021*). Importantly, this finding may provide further insight into the potential mechanisms which may underpin the AFT-induced improvements in sprinting performance in some events. Firstly, the overall stature discrepancy between elite male and female sprinters is approximately 6% (*Weyand & Davis, 2005*), meaning that a similar *absolute* increase in midsole thickness (*e.g.*, the maximum allowed 20 mm) affords a greater *relative* increase in leg length for female sprinters than male sprinters. Given the previously described relationship between leg length and maximal velocity (*Weyand et al., 2022*) and the relationship between stride length and sprinting performance, this may partially explain our observation that female sprinters generally benefit more from AFT than males. Similarly, the geometry of AFT may also influence the sex-specific results observed in this study. Although World Athletics rules stipulate that a marginally thicker sole beyond the 20 mm regulation is permitted in the case of larger shoe sizes (*World Athletics, 2021*), we understand that the 20 mm stack height is not scaled proportionately according to shoe size, and the 20 mm midsole stack is kept relatively consistent across shoe sizes. Theoretically, this creates a more advantageous lever for those with smaller shoe sizes than for those sprinters with larger shoe sizes, due to unique midsole thickness/foot length ratios. Given that continued horizontal force application to the ground at high velocities is a key discriminator of sprint performance between males and females (*Slawinski et al., 2017*), this potential creation of more advantageous levers *via* smaller shoe sizes may help to explain why female sprinters appear to accrue greater benefits from AFT than male sprinters. Finally, differences in body mass may interact with energy restitution and conformity of both the rigid plate and the midsole foam. Combined, these factors may help to explain the sex-specific improvements in performance achieved *via* AFT.

Similarly, but more tentatively, we observed that performance improvements were generally more pronounced in the top 20 compared to the top 100 athletes. While this may be partly explained by statistical factors related to sample size, it may also suggest that AFT preferentially benefits sprinters with certain characteristics, such as technique or specific

strength. Indeed, there is evidence indicating that optimal longitudinal bending stiffness of sprint spikes is specific to the individual (*Stefanyshyn & Fusco, 2004*), and may be mediated such as toe flexor strength, plantar flexor strength, rebound jump performance and body mass (*Willwacher et al., 2016*; *Nagahara, Kanehisa & Fukunaga, 2017*). Similar performance-level dependency has been reported with the AFT-induced enhancement of distance running performance, whereby large variations in the magnitude of performance enhancement have been observed which are partially mediated by the standard of the athlete (*Knopp et al., 2023*). However, given that our top 20 and top 100 cohorts have some overlap, this result should be interpreted with caution, and future studies are needed to more clearly elucidate the potential performance-level specific improvements associated with the use of AFT.

Although we provide initial insight into the recent improvements in some sprint events and the potential performance-enhancing effects of AFT on sprint times, the consistency of our results warrants further discussion. For comparison, studies investigating the influence of AFT on annual long-distance road-race times in elite athletes have reported a universal benefit for all events assessed across both sexes (*Rodrigo-Carranza et al., 2022*, *2021*; *Bermon et al., 2021*). Contrarily, we observed that recent improvements (regardless of AFT influence) were not consistent across all events or across all years, and that AFT was not a significant predictor of performance in four of the ten events analysed. Combined, our data suggests that although AFT in sprint spikes influences performance in some events, they do not discriminate sprint performance to the same extent as AFT discriminates road-racing performance. Some of this inconsistency may be explained by differences in the adoption of AFT in different events and different years. For example, in 2021 only 42.5% of the top 20 athletes (across all events) wore AFT, whereas in 2022, 88.5% of the top 20 athletes utilised AFT. However, our mixed model analyses revealed that other factors are likely also involved in recent sprint time improvements. Changes in factors such as athlete characteristics like age (*Elmenshawy, Machin & Tanaka, 2015*) and stature (*Marck et al., 2017*), weather conditions, career trajectories, changes in training methods and injury status, sex-based and event-based differences in proximity to physiological limits, and increased globalisation are all candidate mediators of performance changes. The COVID-19 pandemic also provided a unique set of circumstances which conceivably influenced the observed performance increases. For example, athletes were afforded the opportunity to train for a prolonged period of time without reducing load as they typically would to peak for major competitions. It is also noteworthy that there was a 46% reduction in drug testing worldwide in 2020 (*World Anti-Doping Authority, 2021*), allowing athletes more opportunity to enhance their performances exogenously (*Negro, Di Trana & Marinelli, 2022*; *Lima et al., 2021*). Given the history and prevalence of performance enhancing drugs in track and field (*Faiss et al., 2020*; *Berthelot et al., 2015*), this is a plausible explanation for some improvement.

There are also limitations of our study which must be considered when interpreting the current results. This is perhaps most practically demonstrated by the unexpected finding of no significant improvements in the men's 400 m hurdles, despite nine of the top 10 times in history being run since the introduction of AFT in 2020. This highlights the limitations

of our initial statistical approach in dealing with outliers, such as Karsten Warholm's 2021 world record (which he ran without AFT), likely due to large standard deviations. The authors do not propose that no meaningful change has occurred in this event, but rather concede the limitations of the initial statistical approach. Interestingly, our mixed model analysis detected that year was a significant predictor of men's 400 m hurdles performance. Further, another weakness of this study was the failure to account for more deterministic factors in the mixed model analysis, such as age, training content and context, environmental conditions and athlete nationality. Importantly, annual changes in competition opportunity have been demonstrated to influence annual performances (*Haake, Foster & James, 2014*). In this specific case, the absence of a major global championship in 2018 may influence the results. Finally, the dataset is limited by size, with only two years of data where athletes had the opportunity to wear AFT. This may limit the interpretation of our mixed model results.

## CONCLUSION

This is the first evidence indicating that sprint times have become significantly faster in some events in the last two years, and that these improvements may be partially driven by technological innovation with sprint footwear design, which aligns with our hypothesis. Further, these improvements appear to be mediated by event, sex and potentially the level of athlete.

Future studies should seek to identify the precise mechanisms through which AFT may improve sprint performance in both sexes independently, and to elucidate the athlete characteristics which may moderate these performance enhancing effects, such as athlete stature, foot-length/midsole thickness ratio, sprinting mechanics and specific strength characteristics. Additional analysis of recent performance trends in events which do not have superspikes available (for example, shot put and discus) would also provide insight into whether recent track performance improvements have been driven by technology or by more general sport-wide improvements in training methodology and competition opportunities, for example. Further, given recent commentary on the potentially enhanced risk of injury with AFT (*Tenforde et al., 2023*), the long-term ramifications of repeated exposure to AFT in sprint spikes should be investigated, especially in youth and developing athletes.

## ACKNOWLEDGEMENTS

The authors wish to sincerely thank Sean Whipp (Whipp Sports) for his assistance with identifying the spikes of athletes.

### Funding

The authors received no funding for this research. We received support for the APC from the German Research Foundation Project No. 512648189 and the Open Access Publication Fund of the Thueringer Universitaets-und Landesbibliothek Jena. The funders had no role

in study design, data collection and analysis, decision to publish, or preparation of the manuscript.

## Grant Disclosures
The following grant information was disclosed by the authors:
APC from the German Research Foundation: 512648189.

## Competing Interests
Astrid Zech is an Academic Editor for PeerJ.

## Author Contributions
- Joel Mason conceived and designed the experiments, performed the experiments, analyzed the data, prepared figures and/or tables, authored or reviewed drafts of the article, and approved the final draft.
- Dominik Niedziela performed the experiments, analyzed the data, prepared figures and/or tables, authored or reviewed drafts of the article, and approved the final draft.
- Jean-Benoit Morin analyzed the data, authored or reviewed drafts of the article, interpreted the data, and approved the final draft.
- Andreas Groll performed the experiments, analyzed the data, prepared figures and/or tables, authored or reviewed drafts of the article, and approved the final draft.
- Astrid Zech conceived and designed the experiments, authored or reviewed drafts of the article, and approved the final draft.

## Human Ethics
The following information was supplied relating to ethical approvals (*i.e.*, approving body and any reference numbers):

The Ethics Commission of the Friedrich Schiller University Jena granted ethical approval to complete this study: FSV 23/057.

## Data Availability
The raw data used for analysis are available in the Supplemental Files.

## Supplemental Information
Supplemental information for this article can be found online at http://dx.doi.org/10.7717/peerj.16433#supplemental-information.

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
