# Peer review of "The potential impact of advanced footwear technology on the recent evolution of elite sprint performances"

_PeerJ, doi:10.7717/peerj.16433_

## Round 0.1 · original submission · Major Revisions

Both reviewers raised important points that I think all need to be addressed in revisions to the manuscript (not just rebuttals to the reviews). In particular:

- Both reviewers had concerns over the seemingly arbitrary choice of 2016 as the starting point for the data. The basis for this choice should be justified and the question of whether categorically different results are achieved if a different starting point is used should be addressed.

- Consistent with Reviewer #2, I was unclear on why paired tests were used. This seems appropriate only if it is the same athletes being compared to themselves, which is presumably not the case unless I misunderstood what was done exactly (e.g. the top 20 athletes in different years are not the same 20 athletes).

·

Basic reporting

Overall, the manuscript was clear, well written and easy to follow. There was one overly long paragraph in the discussion (starting on line 273), which the authors should consider breaking up to aid flow and readability. Referencing was generally good throughout, although the Healey et al (2022) study cited on line 122 and elsewhere was not included within the reference list. Please cross-check the in-text citations with the reference list throughout the manuscript.
It seemed to me that there was some repetition in the data presentation between table 1 and figures 1-3, to the extent that I’m not certain that both the table and the figure are required. Please consider condensing the information contained across these two formats in to one that best summarises the intended overall message. Further, the ordering of events was different between tables 1 & 3, which could potentially cause confusion to the reader. I suggest using a consistent ordering throughout.
All appropriate raw data appears to have been shared, and in my opinion the submission is sufficiently self-contained and represents an appropriate unit of publication.

Experimental design

As far as I am aware, this is original primary research, and it is within the scope of the journal. The aims and hypotheses that are presented are relevant and meaningful, and the research has clear potential to fill an identified knowledge gap.
I have no queries about the ethical standards of the work. I am a little unsure, however, about some of the specifics of the research design, and would appreciate some further clarification here. A major premise of the analysis is that performances in the analysed events have reached a plateau. Whilst a case is made for this within the introduction, and additionally based on the data analysed form 2016-2019, I feel that further explanation and evidence is required. I note that in the supplementary data files, data for the top 100 times have been provided going back to 2010, yet only the data to 2016 have been included in the manuscript. This appears to be something of an arbitrary cut-off, and I wonder if by including the data back to 2010 in the analysis the findings about the timing of the plateau may be different. On what basis was the 2016 cut-off chosen?
Additionally, have the authors considered the potential effect of the type of global competition that is held within each year (Olympics v World Championships), and the differences in qualification that may or may not have some influence on the top times in any given year. Further, some years don’t have a major global championship (e.g. 2018). Please comment on the potential impact of this within your study design.
The introduction section (line 76) mentions the plateau in some women’s events since 1994, but doesn’t mention the potential link to this of the introduction of more routine drug testing not long prior to that date. This point is picked up a little in the discussion, but I feel is also important to raise regarding the initial context of the study.
Overall, I found the description of the methods to contain sufficient information to be reproducible.

Validity of the findings

All necessary underlying data have been provided and appear to be robust. I do have a query regarding the statistical approach, however. On line 169 it is stated that the normal distribution assumption was verified. However, aren’t the data being analysed almost by definition being taken from a very specific part of that normal distribution – i.e. from the very tail at the extreme elite end of performance? This is then represented in the shapes of the curves presented in Figures 1-3. Please clarify.
I felt that in places throughout the manuscript that the findings and conclusions were at times overly confidently or strongly stated. It felt to me that the phrasing from line 260 that “our results provide initial evidence that along with the technological innovation there is meaningful advancement in sprint performances” gave a more appropriate reflection of the likely contribution to the advancement of knowledge of the current study. I suggest that this tone is adopted throughout.
I also felt that some of the explanations for the reasons behind the findings could do with more explanation in a couple of instances; Firstly, the differences in findings between the male and female athletes could potentially do with further explanation – is it possible to provide any theoretical predictions of the likely magnitude of the effects of the mechanisms that you propose for explaining these differences? Secondly, can you comment further on the inconsistency in the findings from 2021 and 2022 compared to pre-ATF? i.e. if the ATF were to have such a clear effect after their adoption, would we expect those effects to be consistent across both years in which they have been worn? Please comment on these points.

Additional comments

The wording on line 118 appears to include some unintended repetition, which makes its meaning hard to follow. Please clarify.

·

Basic reporting

Overall, this paper was a great read and a nice extension to the referenced papers on road running performances and AFT. While the introduction could use a little polishing the content was all there, and a lot of thought was clearly put into the discussion.

Experimental design

Meets all the check boxes.

Validity of the findings

I have one concern about the statistical approach (see below), otherwise robust.

Additional comments

My post-doc Zach Barrons, PhD assisted me with this review.

Overall
The purpose of this paper was to determine if the introduction of AFT spikes have meaningfully impacted sprint event performances and to determine whether they affect one sex mores so than the other.

Overall, this paper was a great read and a nice extension to the referenced paper on road running performances and AFT. While the introduction could use a little polishing the content was all there, and a lot of thought was clearly put into the discussion.

We have several important concerns that we would like to see addressed:

Our main concern is whether the use of paired t-tests is appropriate in this case. There is no reason why data points between years should be considered pairs, and since all the data is ranked (top 20, top 100), this probably makes the use of paired t-tests even less appropriate. Consider the following example. Year 1, top 100: 10.01s – 11.00s; year 2, top 100: 10:00s – 10.99s (let’s say 1 athlete improves from 11:00 to 10:00s). Paired t-test based on rank would suggest a significant improvement (p<0.0000001), as there’s 100 ‘pairs’ where year 2 is 0.01 faster. However, if these are considered 2 independent samples (which they are), these are not significantly different (p=0.8).

Secondly, a rationale should be provided for starting the analyses with the year 2016.

Another concern involves the examination of the top 20 and top 100 performers. As written, for the first half of the paper it appeared to me that you set off from to start to look at these two distinct groups however, it is unclear whether the top 100 analysis includes the top 20 performers as well. Based on the methods I think the focus on 2 groups may have been the result of an inability to identify footwear. Please clarify when you say top 100 vs top 20, does the top 100 contain the top 20? As written, that’s how I would interpret it but that seems like a flawed approach, when aiming to evaluate groups with different performance levels.

Finally, a preprint addressing similar questions using similar methodology is available ( https://sportrxiv.org/index.php/server/preprint/view/297 ) and should be discussed here.

Abstract
Line 32 – We are unsure what you mean by “emphatic performance benefits” perhaps consider the use of a different word.
Line 33 – please change “has” to “have”
Line 35 – Perhaps consider being more specific. It is not clear based on what whether AFT spikes work.
Line 37 – please clarify that you’re looking at improvements in athletes’ performance (I assume)
Line 43 – the structure of this sentence is a little awkward, please consider revising. Specifically “between the years each year for each event”.
Line 47 – please remove “for”

Introduction

Line 65 – “The” can be removed.
Line 71 – please specify the end of the time period referenced or the additional decades in question. “the decades from the 1990s” until when?
Line 72 – please consider rearranging the sentence to state that sprint times are plateauing.
Line 76 – please change “had” to “has”
Line 76 – it might be good to reiterate the changes in drug testing in the early 90s (many would consider the late 80, early 90 women’s sprint performances suspicious)
Line 84 – the shoe characteristics that define AFT are not mentioned until Line 91, and come too late. I can see it would break the flow to add this definition here, but please reorder things, to not leave the reader hanging…
Line 95 – please change “to track spikes” to “in track spikes”
Line 97 – “The” can be removed.
Line 113 – please remove “the” from “the AFT sprint spikes”
Line 115 – please consider changing “AFT to sprint spikes”. Previously you’ve used AFT to describe footwear that incorporate key features and on first read its confusing if you’re talking about the key features or the footwear. Maybe change “AFT to sprint spikes” to “AFT to sprinting”.
Line 117 – please change “throws” to “throwing” or “throw”
Line 118 – please delete “gold medals in athletics throws events”; change “Olympic” to “Olympics”
Line 120 – please consider altering this sentence “Despite…”. The structure of the sentence is a little awkward. Also please change “is” to “ has” on Line 122.
Line 123- aim 1: I’m currently wondering why you specify top 100 and top 20. Are the top 20 not part of the top 100? I think I’m missing something. As specified for line 37, please clarify that you’re looking at improvement in performance of the athletes.
Line 125- please considering changing the line from “the advent and thus extensive use” to “the introduction” (or something similar). I think it may clarify what you mean.
Line 126 – please clarify if you mean all recent performances or recent improvement in performance?
Line 127 – please consider changing “level-specific” to “experience” or something similar to enhance clarity.

Materials and Methods
Database search and data selection
Line 138 – Why did you choose the years 2016 – 2019 as your pre-AFT period? Please consider justifying.
Definition and identification of AFT
Line 153 – please change “uses” to “incorporates” or something similar
Line 155 – would it be possible to provide a list of models identified as AFT in the supplementary materials
Line 156:161 - please consider moving these lines to follow line 147.
Data analysis and statistics
Line 164 – Please consider removing this sentence. I think that goes without saying.
Line 166 – pairwise, see main concern above

Results
Line 194 – since one-sided t-tests were performed, “differences” should be specified as “improvements”

Discussion
Line 251 – When you compared the performances of the top 100 athletes to the top 20, did you remove the top 20 or were they still included in the top 100?
Line 253 – “influencing” is too strong here, given the correlational analysis
Line 259 – please change “distance” to “distances”
Line 273:326 – although the authors should be commended for discussing potential mechanisms explaining their findings. There’s currently too much speculation that is presented with too much confidence. Please add more clearly that this is all speculative based on observations from studies not addressing midsole thickness per se.
Line 282 – alternatively, they might as well create less beneficial lever arms, as for a similar hip torque production a longer effective leg length will result in a smaller propulsive force… basically the opposite of the suggested mechanism… so the same citations can be used for the opposite argument…
Line 293 – observations of reduced leg length resulting in slower speed, do not necessarily mean that increases in leg length result in higher speed…

---

## Round 0.2 · Minor Revisions

A few minor remaining still to address from the reviewers.

·

Basic reporting

In the main, I am satisfied that the authors have addressed my previous comments regarding basic reporting. I respectfully disagree with the rationale presented for including data repeated in both table 1 and figures 1-3, but I don’t feel that there is sufficient need to demand that the authors move either of those formats to the supplementary files unless the academic editor strongly that should be the case.

Experimental design

Again, I am generally satisfied with the changes that have been made in response to my previous suggestions in this section. However, I feel that some of the points that were made within the rebuttal to the editors and reviewers should be included in the manuscript itself. This can be done around lines 144-146, where the citations to the work of Rodrigo-Carranza and colleagues are made. Specifically, reference should be made to your own supplementary data, and to the Willwacher et al. preprint, that both support the case the extending the analysis back to 2010 would have no effect on the analysis.
As a minor follow up point, I also respectfully disagree with the point about the large data set and the difficulties of presenting it being a reason to curtail the analysis. Again, I am not asking for this to be changed within the manuscript, but I feel that the analysis and reporting should be driven by the data and the questions being asked, not by the challenges or complexities of data visualisation, especially with things having advanced so much in that area in recent years.

Validity of the findings

I am happy with the response given by the authors and the changes made to the manuscript as a result in relation to all of my previous comments in this section.

Additional comments

I have no further comments.

·

Basic reporting

All my concerns have been properly addressed

Experimental design

All my concerns have been properly addressed

Validity of the findings

All my concerns have been properly addressed

Additional comments

All my concerns have been properly addressed, two final suggestions:
L78 (tracked changes document) "1896-2008 indicating" change to "1896-2008 indicate"
L339 (tracked changes document) "which do not directly investigate midsole thickness" consider briefly discussing the findings of Barrons et al (https://doi.org/10.1080/19424280.2023.2218321) here, who observed no improvements in running economy from increased midsole thickness.

---

## Author Rebuttal · Round 0.2

# Response to reviewer comments
*The potential impact of advanced footwear technology on the recent evolution of elite sprint times*

Dear editor and reviewers,

Thanks for the opportunity to submit a revision of our manuscript entitled "The potential impact of advanced footwear technology on the recent evolution of elite sprint times (87409)".

Thanks to all involved for their time and expertise. We've made a series of changes to our manuscript based on the constructive feedback received, and believe the manuscript is much improved as a result. Please see below for responses to specific comments, and see the attached manuscript for changes.

## Editor
**Both reviewers raised important points that I think all need to be addressed in revisions to the manuscript (not just rebuttals to the reviews). In particular:**

**- Both reviewers had concerns over the seemingly arbitrary choice of 2016 as the starting point for the data. The basis for this choice should be justified and the question of whether categorically different results are achieved if a different starting point is used should be addressed.**

Our work was heavily influenced by two seminal papers by Víctor Rodrigo-Carranza et al. which identified the influence of AFT on improvements in road-racing times, published in high quality journals such as Scientific Reports. In both of these papers, Rodrigo-Carranza et al. used a period of four years to answer a very similar question to ours. Therefore, our use of a six-year period to determine the most recent evolution of sprint performances is in line with previous work.

Further, our ambition was to characterise the improvements in sprint times compared to the most recent sprint performances. There is no categorical difference in results if we extend the starting date back to 2010. To confirm this, we draw your attention to the recent pre-print of Willwacher et al. (2023), who reported very similar findings to us using the period 2010-2023 using similar analytical approach to the first part of our analysis. We would also like to highlight that our paper extends comprehensively on Willwacher et al's approach, by actually providing a link between superspike use and performance improvements rather than just pointing out recent trends in performance times. Our regression analysis uses a period of four years without the superspikes and two years with the superspikes to highlight that the superspikes are associated with faster times in most sprint events, which is the key finding of our paper.

An additional key factor in our decision to start in 2016 was maximising the clarity of our communication of results. Because our results are already considerable in depth (three different analyses) and breadth (ten different events), we made the decision to choose a relatively recent cut-off point to contain the scope of the study so our readers can have clarity on the main findings. In the current format, some of our results tables already extend a full A4 page. If we extended our results to 2010, the logistics of creating multi-page tables and multi-page graphs would dramatically alter the clarity of results for the reader.

In sum, given 1) the precedent for using this length of time period in this very specific discipline of research, 2) our main objective to look at the most recent evolutions in performance and link the

use of AFT to changes in times, 3) the evidence that there is no categorical difference in our results by changing the cut-off point, 4) and the clarity that using 2016 affords us in communicating the key results effectively, we are confident that we have made the right decision to use 2016 as a cut-off point.

We have also added a brief statement in the manuscript regarding this: "We selected 2016 as a cut-off point to capture the most recent evolution in sprint performances, in line with time periods used by previous studies characterising the influence of AFT on road-racing times (Rodrigo-Carranza et al. 2021; Rodrigo-Carranza et al. 2022)" (line 144).

*Rodrigo-Carranza, V., González-Mohíno, F., Santos del Cerro, J., Santos-Concejero, J. and González-Ravé, J.M., 2021. Influence of advanced shoe technology on the top 100 annual performances in men's marathon from 2015 to 2019. Scientific reports, 11(1), p.22458.*

*Rodrigo-Carranza, V., González-Mohíno, F., Santos-Concejero, J. and González-Ravé, J.M., 2022. Impact of advanced footwear technology on elite men's in the evolution of road race performance. Journal of Sports Sciences, 40(23), pp.2661-2668.*

*Willwacher, S., Mai, P., Helwig, J., Hipper, M., Utku, B. and Robbin, J., Does advanced footwear technology improve track and road racing performance? An explorative analysis based on the 100 best yearly performances in the world between 2010 and 2022: The effects of super-footwear on world-class track and road racing performances. (Pre-print)*

**Consistent with Reviewer #2, I was unclear on why paired tests were used. This seems appropriate only if it is the same athletes being compared to themselves, which is presumably not the case unless I misunderstood what was done exactly (e.g. the top 20 athletes in different years are not the same 20 athletes).**

Our apologies for this misunderstanding. As described in the response to Dr Hoogkamer below, we used independent/unpaired t-tests for our analysis. "Pairwise" t-tests simply refers to the function in R-Studio which describes a multiple t-test approach that checks for differences between all possible combinations of groups (in this case, year = group). In this instance, "pairwise" is unrelated to whether the test is paired or unpaired. The actual t-tests we used were unpaired/independent, in line with the suggestions from Dr Hoogkamer. We acknowledge that we allowed too much opportunity for misinterpretation here and accept full responsibility for this error. The manuscript has since been updated to provide more clarity.

**Reviewer 1**
**Reviewer: Ian Bezodis**
Thank you Dr Bezodis for your time and expertise. Your feedback has helped us make important clarifications, assisted our decision making for more effective communication of our data, and ultimately improved our manuscript. Please see below for specific responses to your comments.

**Basic reporting**
Overall, the manuscript was clear, well written and easy to follow. There was one overly long paragraph in the discussion (starting on line 273), which the authors should consider breaking up to aid flow and readability.

We have broken this paragraph up to improve readability.

Referencing was generally good throughout, although the Healey et al (2022) study cited on line 122 and elsewhere was not included within the reference list. Please cross-check the in-text citations with the reference list throughout the manuscript.

Thanks for your attention to detail. Our references have been updated accordingly.

It seemed to me that there was some repetition in the data presentation between table 1 and figures 1-3, to the extent that I'm not certain that both the table and the figure are required. Please consider condensing the information contained across these two formats in to one that best summarises the intended overall message.

Our authorship group had the same debate pre-submission, and ultimately decided to include both the tables and figures for the following reason: we hope that our manuscript will be widely read beyond academia, including by coaches, athletes and those with a general interest in trends in sport performance. We believe that the tables and graphs will distinctly satisfy the unique needs of different readers. For example, we believe that coaches will look for specific relatable times within the tables to use as familiar reference points, whereas those generally interested in the trend will simply scan the results for a snapshot of the general direction; in which case the graphs serve as the most appropriate format. Considering the unique purposes they serve, we hope that you ultimately agree with our decision to include tables and graphs. However, if the reviewers and editor all disagree with our decision, we can move either the tables or the figures to the supplementary files.

All appropriate raw data appears to have been shared, and in my opinion the submission is sufficiently self-contained and represents an appropriate unit of publication. Further, the ordering of events was different between tables 1 & 3, which could potentially cause confusion to the reader. I suggest using a consistent ordering throughout.

Thanks for your attention to detail. The ordering of information in our tables has been updated for consistency.

**Experimental design**
As far as I am aware, this is original primary research, and it is within the scope of the journal. The aims and hypotheses that are presented are relevant and meaningful, and the research has clear potential to fill an identified knowledge gap.

I have no queries about the ethical standards of the work. I am a little unsure, however, about some of the specifics of the research design, and would appreciate some further clarification here. A major premise of the analysis is that performances in the analysed events have reached a plateau. Whilst a case is made for this within the introduction, and additionally based on the data analysed form 2016-2019, I feel that further explanation and evidence is required. I note that in the supplementary data files, data for the top 100 times have been provided going back to 2010, yet only the data to 2016 have been included in the manuscript. This appears to be something of an arbitrary cut-off, and I wonder if by including the data back to 2010 in the analysis the findings about the timing of the plateau may be different. On what basis was the 2016 cut-off chosen?

Regarding this point, we repeat the response to the editor above:

Our work was heavily influenced by two seminal papers by Víctor Rodrigo-Carranza et al. which identified the influence of AFT on improvements in road-racing times, published in high quality

journals such as Scientific Reports. In both of these papers, Rodrigo-Carranza et al. used a period of four years to answer a very similar question to ours. Therefore, our use of a six-year period to determine the most recent evolution of sprint performances is in line with previous work.

Our ambition was to characterise the improvements in sprint times compared to the most recent sprint performances. There is no categorical difference in results if we extend the starting date back to 2010. To confirm this, we draw your attention to the recent pre-print of Willwacher et al. (2023), who reported very similar findings to us using the period 2010-2023 and a similar analytical approach. We would also like to highlight that our paper extends comprehensively on Willwacher et al's approach, by actually providing a link between superspike use and performance improvements rather than just pointing out recent trends in performance times. Our regression analysis uses a period of four years without the superspikes and two years with the superspikes to highlight that the superspikes are associated with faster times in most sprint events, which is the key finding of our paper.

An additional key factor in our decision to start in 2016 was maximising the clarity of our communication of results. Because our results are already considerable in depth (three different analyses) and breadth (ten different events), we made the decision to choose a relatively recent cut-off point to contain the scope of the study so our readers can have clarity on the main findings. In the current format, some of our results tables already extend a full A4 page. If we extended our results to 2010, the logistics of creating multi-page tables and multi-page graphs would dramatically alter the clarity of results for the reader.

In sum, given 1) the precedent for using this length of time period in this very specific discipline of research, 2) our main objective to look at the most recent evolutions in performance and link the use of AFT to changes in times, 3) the evidence that there is no categorical difference in our results by changing the cut-off point, 4) and the clarity that using 2016 affords us in communicating the key results effectively, we are confident that we have made the right decision to use 2016 as a cut-off point.

We have also added a brief statement in the manuscript regarding this: "We selected 2016 as a cut-off point to capture the most recent evolution in sprint performances, in line with time periods used by previous studies characterising the influence of AFT on road-racing times (Rodrigo-Carranza et al. 2021; Rodrigo-Carranza et al. 2022)", (line 144).

*Rodrigo-Carranza, V., González-Mohíno, F., Santos del Cerro, J., Santos-Concejero, J. and González-Ravé, J.M., 2021. Influence of advanced shoe technology on the top 100 annual performances in men's marathon from 2015 to 2019. Scientific reports, 11(1), p.22458.*

*Rodrigo-Carranza, V., González-Mohíno, F., Santos-Concejero, J. and González-Ravé, J.M., 2022. Impact of advanced footwear technology on elite men's in the evolution of road race performance. Journal of Sports Sciences, 40(23), pp.2661-2668.*

*Willwacher, S., Mai, P., Helwig, J., Hipper, M., Utku, B. and Robbin, J., Does advanced footwear technology improve track and road racing performance? An explorative analysis based on the 100 best yearly performances in the world between 2010 and 2022: The effects of super-footwear on world-class track and road racing performances. (Pre-print)*

Additionally, have the authors considered the potential effect of the type of global competition that is held within each year (Olympics v World Championships), and the differences in qualification that may or may not have some influence on the top times in any given year. Further, some years don't have a major global championship (e.g. 2018). Please comment on the potential impact of this within your study design.

We agree that differences in competition opportunities between years may have additionally influenced our results, and have since added this to our discussion section to aid in the interpretation. Specifically, "Importantly, annual changes in competition opportunity have been demonstrated to influence annual performances (Haake, James & Foster 2013). In this specific case, the absence of a major global championship in 2018 may influence the results". (Lines 404-407).

**The introduction section (line 76) mentions the plateau in some women's events since 1994, but doesn't mention the potential link to this of the introduction of more routine drug testing not long prior to that date. This point is picked up a little in the discussion, but I feel is also important to raise regarding the initial context of the study.**

Good point. We've updated our introduction accordingly, which now includes the following addition:

"One model incorporating performances from 1896-2008 indicating that no meaningful progression has occurred in 4 out of 5 women's sprint events since 1994 (Berthelot et al. 2010), which may be partially explained by the introduction of routine performance enhancing drug testing (Haake, Foster & James 2014) (line 77).

**Overall, I found the description of the methods to contain sufficient information to be reproducible.**

**Validity of the findings**
All necessary underlying data have been provided and appear to be robust. I do have a query regarding the statistical approach, however. On line 169 it is stated that the normal distribution assumption was verified. However, aren't the data being analysed almost by definition being taken from a very specific part of that normal distribution – i.e. from the very tail at the extreme elite end of performance? This is then represented in the shapes of the curves presented in Figures 1-3. Please clarify.

Thanks for highlighting this. For some events, the validity of normality assumption is indeed somewhat critical, particularly because the underlying top 20 or top 100 performance variables are cut off at the upper tails. Despite this limitation, t-tests remain the most robust tool for testing our initial hypothesis over non-parametric tests. However, we certainly see your point and agree that additional clarification is needed. We have consequently decided to validate our findings and selection of analysis, and support our overall findings, by additionally including Wilcoxon-Mann-Whitney tests. With this additional analysis, we tested for the null hypothesis that it is equally likely that a value chosen at random from one year is greater or less than a value chosen at random from another year's population. The corresponding results confirm our general results from the initial t-test analysis. For example, in the top 100 analysis, there are no significant differences between years 2016-2019 in any event, and all events displaying significant effects between pre-AFT years and AFT years with the initial t-test analysis also display significant differences with the Wilcoxon-Mann-Whitney tests. We have included the results of this additional testing as a supplementary file (supp file 3), and have added a small section to the manuscript to acknowledge this and to clarify our statistical approach.

Further, for the linear mixed models that we fit on the data of the top 20 athletes, the normality assumption is implicitly relevant. However, on the basis of a careful goodness-of-fit analysis

using residual and QQ-plots, we found no mentionable violations here for all fitted models. This is now also mentioned in the manuscript.

Following these clarifications, combined with the fact that our co-author is a professor of statistics, we are confident that our manuscript is now clearer, and our approach is more robust. Thanks to Dr Bezodis for prompting this update.

I felt that in places throughout the manuscript that the findings and conclusions were at times overly confidently or strongly stated. It felt to me that the phrasing from line 260 that "our results provide initial evidence that along with the technological innovation there is meaningful advancement in sprint performances" gave a more appropriate reflection of the likely contribution to the advancement of knowledge of the current study. I suggest that this tone is adopted throughout.

We have updated sections of the discussion in order to provide a more appropriate reflection of how our results contribute to knowledge advancement, and have generally toned down some of our statements regarding the certainty and strength of our findings.

I also felt that some of the explanations for the reasons behind the findings could do with more explanation in a couple of instances; Firstly, the differences in findings between the male and female athletes could potentially do with further explanation – is it possible to provide any theoretical predictions of the likely magnitude of the effects of the mechanisms that you propose for explaining these differences?

We have done our best to speculate on the potential reasons for the differences in performance enhancement differences between males and females, including information on differences in relative leg length (lines 331-334) and differences in shoe geometry as a result of shoe size (lines 341-346). We've since added information related to how differences in body mass may also play a role in energy restitution afforded by the plate and foam combined (lines 346-349).

Unfortunately, the current body of literature does not allow us to confidently theoretically predict the likely magnitude of effects of each of the mechanisms. We feel that this section of our discussion is already quite speculative (as mentioned by the second reviewer), and so we would like to avoid adding additional speculation on top of this.

In future studies, we aim to elucidate some of these mechanisms underpinning the sex-specific differences in performance enhancement. We have accordingly added this to our conclusion section, where we recommend that "Future studies should seek to identify the precise mechanisms through which AFT may improve sprint performance in both sexes independently, and to elucidate the athlete characteristics which may moderate these performance enhancing effects, such as athlete stature, foot-length/midsole thickness ratio, sprinting mechanics and specific strength characteristics".

Secondly, can you comment further on the inconsistency in the findings from 2021 and 2022 compared to pre-ATF? i.e. if the ATF were to have such a clear effect after their adoption, would we expect those effects to be consistent across both years in which they have been worn? Please comment on these points.

Great point. One key factor possibly underlying the inconsistency is the adoption of AFT in 2021 vs 2022. In 2021, approximately 42.5% of the top 20 athletes (across all events) wore AFT, whereas in 2022 88.5% of the top 20 athletes wore AFT. We have now highlighted this in our discussion section (lines 376-379). Our mixed model analysis also indicates that factors outside of the superspikes are responsible for change in improvements, which we've highlighted via the following statement: "Further, our mixed model analyses revealed that other factors are likely involved in recent sprint

time evolutions. Changes in factors such as athlete characteristics like age (Elmenshawy, Machin & Tanaka 2015) and stature (Marck et al. 2017), weather conditions, career trajectories, changes in training methods and injury status, sex-based and event-based differences in proximity to physiological limits, and increased globalisation are all candidate mediators of performance changes" (lines 380-384).

**Additional comments**
The wording on line 118 appears to include some unintended repetition, which makes its meaning hard to follow. Please clarify.

Thanks for your attention to detail. This sentence has been updated accordingly.

**Reviewer 2: Wouter Hoogkamer**
**Basic reporting**
Overall, this paper was a great read and a nice extension to the referenced papers on road running performances and AFT. While the introduction could use a little polishing the content was all there, and a lot of thought was clearly put into the discussion.

**Experimental design**
Meets all the check boxes.

**Validity of the findings**
I have one concern about the statistical approach (see below), otherwise robust.

**Additional comments**
My post-doc Zach Barrons, PhD assisted me with this review.

**Overall**
The purpose of this paper was to determine if the introduction of AFT spikes have meaningfully impacted sprint event performances and to determine whether they affect one sex mores so than the other.

Overall, this paper was a great read and a nice extension to the referenced paper on road running performances and AFT. While the introduction could use a little polishing the content was all there, and a lot of thought was clearly put into the discussion.

Firstly, thank you Dr Hoogkamer (and Dr Barrons) for taking the time to provide your expertise. It's great to receive constructive feedback from a leader in the field of AFT. Your feedback has helped us enhance the quality of our work and provide more clarity with the communication of our results.

We have several important concerns that we would like to see addressed:

Our main concern is whether the use of paired t-tests is appropriate in this case. There is no reason why data points between years should be considered pairs, and since all the data is ranked (top 20, top 100), this probably makes the use of paired t-tests even less appropriate. Consider the following example. Year 1, top 100: 10.01s – 11.00s; year 2, top 100: 10:00s – 10.99s (let's say 1 athlete improves from 11:00 to 10:00s). Paired t-test based on rank would suggest a significant improvement (p<0.0000001), as there's 100 'pairs' where year 2 is 0.01 faster. However, if these are considered 2 independent samples (which they are), these are not significantly different (p=0.8).

Thanks very much for bringing this to our attention. This is purely a communication error on our behalf. We performed unpaired/independent t-tests, not paired t-tests. In this instance, "pairwise" t-tests simply refers to the function in R-Studio which describes a multiple t-test approach that checks for differences between all possible combinations of groups (in this case, year = group). In this case, it is unrelated to whether the test is paired or unpaired. The actual t-tests we used were unpaired/independent. We acknowledge that we've certainly provided ample opportunity for misinterpretation here and accept full responsibility for this. We've accordingly changed the language throughout the manuscript (including tables) to more clearly reflect the statistical testing used.

Secondly, a rationale should be provided for starting the analyses with the year 2016.
Please see the above response to the editor regarding this point. We've updated the manuscript to provide a brief justification for using this time period: *"We selected 2016 as a cut-off point to capture the most recent evolution in sprint performances, in line with time periods used by previous studies characterising the influence of AFT on road-racing times (Rodrigo-Carranza et al. 2021; Rodrigo-Carranza et al. 2022)" (lines 144-146).*

Another concern involves the examination of the top 20 and top 100 performers. As written, for the first half of the paper it appeared to me that you set off from to start to look at these two distinct groups however, it is unclear whether the top 100 analysis includes the top 20 performers as well. Based on the methods I think the focus on 2 groups may have been the result of an inability to identify footwear. Please clarify when you say top 100 vs top 20, does the top 100 contain the top 20? As written, that's how I would interpret it but that seems like a flawed approach, when aiming to evaluate groups with different performance levels.

We can confirm the top 100 includes also the top 20, and therefore the groups are overlapping. We see your point, and we agree that this is a somewhat flawed approach which may detract from our main thesis and findings. This is not a primary feature of our analysis, and thus we've removed it from the introduction (where it was previously listed as a secondary aim of the study) so that the reader is able to focus on the main findings which are reported with more robust analyses.

You are also correct in that the top 20 year-to-year analysis is primarily intended to provide the basis for our subsequent sub-sample analysis in the regressions, where we identified the spikes worn by the top 20 athletes only.

Although we still briefly mention the larger effect sizes in the top 20 compared to the top 100 athletes in the discussion, we have softened this paragraph to more transparently point out the flaws of this approach and further highlighted that the findings should therefore be interpreted with caution. This includes the addition of the following statement: "However, given that our top 20 and top 100 cohorts have some overlap, this result should be interpreted with caution, and future studies are needed to more clearly elucidate the potential performance-level specific improvements associated with the use of AFT" (lines 361-364).

**Finally, a preprint addressing similar questions using similar methodology is available ( https://sportrxiv.org/index.php/server/preprint/view/297 ) and should be discussed here.**

Thanks for this. We became aware of the pre-print after our initial submission, so we're glad to have the opportunity to include it now. We've now mentioned it in both our introduction and discussion sections.

Abstract

Line 32 – We are unsure what you mean by "emphatic performance benefits" perhaps consider the use of a different word.

We have changed "emphatic performance benefits" to "widely-reported benefits". (Line 32)

Line 33 – please change "has" to "have"

Thanks for your attention to detail. Updated accordingly.

Line 35 – Perhaps consider being more specific. It is not clear based on what whether AFT spikes work.

We have updated this sentence accordingly, and it now reads: However, it is not yet clear based on current evidence whether there have been subsequent improvements in sprint performance.

Line 37 – please clarify that you're looking at improvements in athletes' performance (I assume)
Updated accordingly, and now reads "Therefore, the aims of this study were to establish if there have been recent year-to-year improvements in the times of the annual top 100 and top 20 athletes in the men's and women's sprint events".

Line 43 – the structure of this sentence is a little awkward, please consider revising. Specifically "between the years each year for each event".
Sentence has been updated for clarity, and now reads "...t-tests with Holm corrections were performed using the season's best performance of the top 100 and top 20 athletes in each year to identify significant differences between years for each event".

Line 47 – please remove "for"
Updated accordingly.

**Introduction**
Line 65 – "The" can be removed.
Updated accordingly.

Line 71 – please specify the end of the time period referenced or the additional decades in question. "the decades from the 1990s" until when?

Updated accordingly. Sentence now reads: "Following a century of progress, sprint times have somewhat plateaued since the 1990s across most elite sprint events as performances have approached their asymptotic limits (Berthelot et al. 2010, Berthelot et al. 2015, Weiss et al. 2016, Ganse & Degens 2021) (lines 71-73).

Line 72 – please consider rearranging the sentence to state that sprint times are plateauing.

We've updated this sentence accordingly. It now reads:
"Following a century of progress, sprint times have somewhat plateaued since the 1990s across most elite sprint events as performances have approached their asymptotic limits (Berthelot et al. 2010, Berthelot et al. 2015, Weiss et al. 2016, Ganse & Degens 2021) (lines 71-73).

Line 76 – please change "had" to "has"
Updated accordingly.

Line 76 – it might be good to reiterate the changes in drug testing in the early 90s (many would consider the late 80, early 90 women's sprint performances suspicious)

We agree and have updated this section accordingly. It now reads: "One model incorporating performances from 1896-2008 indicating that no meaningful progression has occurred in 4 out of 5 women's sprint events since 1994 (Berthelot et al. 2010), which may be partially explained by the introduction of routine performance enhancing drug testing (Haake, Foster & James 2014)"..

Line 84 – the shoe characteristics that define AFT are not mentioned until Line 91, and come too late. I can see it would break the flow to add this definition here, but please reorder things, to not leave the reader hanging…

This paragraph has been restructured accordingly, and we now define AFT earlier to avoid leaving the reader hanging.

Line 95 – please change "to track spikes" to "in track spikes"
Updated accordingly.

Line 97 – "The" can be removed.
Updated accordingly.

Line 113 – please remove "the" from "the AFT sprint spikes"
Updated accordingly.

Line 115 – please consider changing "AFT to sprint spikes". Previously you've used AFT to describe footwear that incorporate key features and on first read its confusing if you're talking about the key features or the footwear. Maybe change "AFT to sprint spikes" to "AFT to sprinting".
Good point! Updated accordingly.

Line 117 – please change "throws" to "throwing" or "throw"

Updated to "throwing" accordingly.

Line 118 – please delete "gold medals in athletics throws events"; change "Olympic" to "Olympics"
Updated accordingly.

Line 120 – please consider altering this sentence "Despite…". The structure of the sentence is a little awkward. Also please change "is" to " has" on Line 122.
This passage has been restructured for clarity. Line 122 also updated accordingly.

Line 123- aim 1: I'm currently wondering why you specify top 100 and top 20. Are the top 20 not part of the top 100? I think I'm missing something.

Please see our previous comment. We agree, and have consequently amended the manuscript.

As specified for line 37, please clarify that you're looking at improvement in performance of the athletes.

Manuscript updated accordingly.

Line 125- please considering changing the line from "the advent and thus extensive use" to "the introduction" (or something similar). I think it may clarify what you mean.
Updated accordingly.

Line 126 – please clarify if you mean all recent performances or recent improvement in performance?
We have updated this sentence for clarity: "...to establish if there is an association between the introduction of AFT and the potential recent improvements in sprint performances in each event".

Line 127 – please consider changing "level-specific" to "experience" or something similar to enhance clarity.
As per the feedback above, we've removed this from our introduction section.

**Materials and Methods**
Database search and data selection
Line 138 – Why did you choose the years 2016 – 2019 as your pre-AFT period? Please consider justifying.

Please see our previous response regarding this point. We've also now justified this in the manuscript via the following statement:

"We selected 2016 as a cut-off point to capture the most recent evolution in sprint performances, in line with time periods used by previous studies characterising the influence of AFT on road-racing times (Rodrigo-Carranza et al. 2021; Rodrigo-Carranza et al. 2022)" (line 144-146).

Definition and identification of AFT
Line 153 – please change "uses" to "incorporates" or something similar
Updated accordingly.

Line 155 – would it be possible to provide a list of models identified as AFT in the supplementary materials
Nice idea. We've added a list of models identified as AFT in the supplementary materials.

Line 156:161 - please consider moving these lines to follow line 147.
Updated accordingly.

**Data analysis and statistics**

Line 164 – Please consider removing this sentence. I think that goes without saying.
Updated accordingly.

Line 166 – pairwise, see main concern above
Please see our response to the main concern above.

**Results**
Line 194 – since one-sided t-tests were performed, "differences" should be specified as "improvements"
Updated accordingly here and in line 213 (formerly line 216).

**Discussion**
Line 251 – When you compared the performances of the top 100 athletes to the top 20, did you remove the top 20 or were they still included in the top 100?

Please note our comment above regarding this point.

Line 253 – "influencing" is too strong here, given the correlational analysis

We agree. We've consequently softened this statement, and it now reads: "the use of AFT may partially explain these recent improvements in sprint times, with a significant relationship identified in six out of ten events".

Line 259 – please change "distance" to "distances"

Updated accordingly.

Line 273:326 – although the authors should be commended for discussing potential mechanisms explaining their findings. There's currently too much speculation that is presented with too much confidence. Please add more clearly that this is all speculative based on observations from studies not addressing midsole thickness per se.

We agree with your observation and have softened our discussion as a result. Specifically, we now explicitly highlight our speculation regarding mechanisms via the following statement:

"It should be noted that these mechanisms remain purely speculative at this stage, based on studies which do not directly investigate midsole thickness" (lines 315-321).

We've also made minor amendments to some statements throughout the discussion to be more generally hesitant in order to more accurately represent the nature of our speculation.

Line 282 – alternatively, they might as well create less beneficial lever arms, as for a similar hip torque production a longer effective leg length will result in a smaller propulsive force… basically the opposite of the suggested mechanism… so the same citations can be used for the opposite argument…

However, it should be noted that there is also the possibility of creating less beneficial lever arms, as for a similar hip torque production a longer effective leg length will result in a smaller propulsive force. Therefore, studies should seek to clarify these mechanisms and how they ultimately contribute to enhanced performance.

Good point. We've added this to the manuscript and have suggested that future studies seek to elucidate the precise mechanisms via which AFT may enhance sprint performance. This can be found in lines 317-321, and reads: "It should be noted that these mechanisms remain primarily speculative at this stage, based on studies which do not directly investigate midsole thickness. A key example which demonstrates the uncertainty of the mechanisms is that AFT potentially also creates a less beneficial lever arm, considering that for a constant hip torque, a longer effective leg length will result in a smaller propulsive force. Therefore, future studies should seek to clarify the precise mechanisms through which AFT ultimately contributes to enhanced sprint performance".

Line 293 – observations of reduced leg length resulting in slower speed, do not necessarily mean that increases in leg length result in higher speed…

We've updated this statement to include the following: "Although reduced leg length in amputee athletes resulting in slower speed does not guarantee that increasing leg length results in higher speed in able-bodied athletes" (lines 309-311).

---

## Round 0.3 · accepted · Accept

The authors have addressed all of the reviewers' comments and the manuscript seems ready for publication.